# The Structural, Electronic, and Optical Properties of Ge/Si Quantum Wells: Lasing at a Wavelength of 1550 nm

**DOI:** 10.3390/nano10051006

**Published:** 2020-05-25

**Authors:** Hongqiang Li, Jianing Wang, Jinjun Bai, Shanshan Zhang, Sai Zhang, Yaqiang Sun, Qianzhi Dou, Mingjun Ding, Youxi Wang, Dan Qu, Jilin Du, Chunxiao Tang, Enbang Li, Joan Daniel Prades

**Affiliations:** 1Tianjin Key Laboratory of Optoelectronic Detection Technology and Systems, School of Electronics and Information Engineering, Tiangong University, Tianjin 300387, China; 1830091211@tiangong.edu.cn (J.W.); baijinjun@gmail.com (J.B.); zhangshanshan@tiangong.edu.cn (S.Z.); 1931075513@tiangong.edu.cn (S.Z.); 1931075519@tiangong.edu.cn (Y.S.); douqianzhi@126.com (Q.D.); dingmingjun@tiangong.edu.cn (M.D.); wangyouxi@tiangong.edu.cn (Y.W.); jackieqd163@163.com (D.Q.); 1831095429@tiangong.edu.cn (J.D.); tangyifei82@hotmail.com (C.T.); 2Tianjin Key Laboratory of Optoelectronic Sensor and Sensing Network Technology, Institute of Modern Optics, Nankai University, Tianjin 300071, China; 3Centre for Medical Radiation Physics, University of Wollongong, Wollongong, NSW 2522, Australia; enbang_li@uow.edu.au; 4MIND, Departament of Electronics and Biomedical Engineering, Universitat de Barcelona (UB), E-08028 Barcelona, Spain; 5Institute of Nanoscience and Nanotechnology (IN ^2^UB), Universitat de Barcelona (UB), E-08028 Barcelona, Spain

**Keywords:** Ge/Si quantum well, tensile strain, n-type doping, lattice thermal mismatch process, Si-based light source

## Abstract

The realization of a fully integrated group IV electrically driven laser at room temperature is an essential issue to be solved. We introduced a novel group IV side-emitting laser at a wavelength of 1550 nm based on a 3-layer Ge/Si quantum well (QW). By designing this scheme, we showed that the structural, electronic, and optical properties are excited for lasing at 1550 nm. The preliminary results show that the device can produce a good light spot shape convenient for direct coupling with the waveguide and single-mode light emission. The laser luminous power can reach up to 2.32 mW at a wavelength of 1550 nm with a 300-mA current. Moreover, at room temperature (300 K), the laser can maintain maximum light power and an ideal wavelength (1550 nm). Thus, this study provides a novel approach to reliable, efficient electrically pumped silicon-based lasers.

## 1. Introduction

The photonic integrated circuit is an emerging technology that uses a crystalline semiconductor wafer as the platform for the integration of active and passive photonic circuits along with electronic components on a single microchip. Silicon photonics is the platform of choice due to its scalability, low cost and functional integration. With the necessary expertise, this technology will continue to enable innovative solutions utilizing silicon optical circuits and micro-optics while allowing for the optimal integration of control electronics and system packaging. Over the past two decades, silicon photonics has grown from academic research into an industrially viable technology, largely driven by applications in data communication. We have seen that the material palette of silicon photonics extends well beyond silicon to incorporate low-loss waveguides, efficient modulators and detectors and to produce optical gain. Apart from silicon-based light sources, other silicon-based photonic devices, such as waveguides [1,2], gratings [3,4], modulators [5,6], detectors [7,8], polarization splitters [9], mixers and filters [10], have been realized. Nevertheless, the realization of a silicon-based light source remains an urgent issue in the field of silicon-based photonic devices. The difficulty is that the common group III-V compound semiconductor laser is not compatible with the silicon-based complementary metal-oxide-semiconductor (CMOS) process [11,12]. Pure group IV semiconductor materials are compatible with the silicon-based CMOS process; however, group IV element semiconductors are indirect-bandgap semiconductors, and when used as the active region for a laser, their emission efficiency is very low. Regardless, the development of silicon-based light sources has not been interrupted, as evidenced by advances including pumped Raman lasers [13], Er-doped Si using plasma-enhanced chemical vapor deposition (CVD) [14], Si nanocrystals [15], and group III-V materials grown on Si [11,12,16].

Ge, which is a group IV material, is compatible with Si technology and is a quasi-direct-bandgap material with a 0.13-eV gap between the direct and indirect bands. In addition, tensile strain can be applied to Ge to eliminate the bandgap between the direct and indirect bands and to increase the efficiency of the direct-band transition. According to the Van de Walle deformation potential theory [17], when tensile strain is applied, the direct band and indirect band of tensile-strained Ge will decrease simultaneously; otherwise, the direct band will decrease more than the indirect band, which means that the radiative recombination of the direct band will improve. In fact, when a 2% tensile strain is introduced to Ge, Ge becomes a direct-band semiconductor [18,19]. In our opinion, two methods are usually applied to realize tensile strain in Ge. The first method involves a Ge/Sn-obtained superlattice [20,21]. Because the lattice constant of Sn is larger than that of Ge, Ge can undergo tensile strain due to the Sn-obtained layer. The second method involves thermal lattice mismatch technology applied to Ge and Si. Since the thermal expansion coefficient of Ge is greater than that of Si, tensile strain can be introduced in Ge on a Si layer [22,23,24]. In recent research, a 0.2% tensile strain in the Ge of Ge p-i-n photodetectors on a Si platform was realized with thermal mismatch technology [25]. Moreover, when n-type doping is introduced to Ge, the *L* valley of the Ge band becomes electronically filled. Therefore, the bandgap between the direct band and the indirect band of Ge can be compensated for by n-type doping [26]. Researchers at MIT used ultrahigh-vacuum CVD to obtain a high active carrier concentration in n-type Ge [27]. Another particular characteristic of Ge is that the direct bandgap is 0.8 eV, which means that the radiative recombination of the direct band corresponds to a wavelength of 1550 nm, which is the telecom wavelength. Recently, researchers have been conducted to using Ge quantum structures to realize the C-band telecom wavelength emission [28,29,30]. However, certain defects persist in these devices, such as a low luminous efficiency, low luminous power and imperfect device structure optimization. We urgently need a laser that is compatible with the silicon-based CMOS process and whose performance is comparable to that of the traditional group III-V compound semiconductor laser.

In this paper, a group IV side-emitting laser based on a 3-layer Ge/Si QW is proposed. We discussed the selection of the thickness and doping concentration of the QW. Considering the spontaneous thermal characteristics of the device, we also designed a unilateral negative electrode structure to provide the device with enough heat dissipation space. We further described the selection of the cavity length to realize single-longitudinal-mode emission and to maximize the power output. We introduced a 0.2% tensile strain into the Ge layers using a lattice thermal mismatch process and 1 × 10${}^{19}$ cm${}^{-3}$ n-type doping using the thermal diffusion process to improve the direct-band radiative recombination in Ge. The results show that the laser exhibits good edge emission characteristics and a large laser emission power at room temperature. We also discussed the effect to the performance of lasers. It is found that increasing the number of QWs can improve the luminous power of laser, but it will lead to more significant carrier loss and decreased gain.

## 2. Device Design and Characterization

The changes in the Ge band under tensile strain and n-type doping are shown in Figure 1. Figure 1a depicts the band structure of an unstrained Ge. The bandgap of *L* (indirect band) is 0.644 eV, and that of Γ (direct band) is 0.8 eV. Ge is an indirect-band semiconductor that cannot achieve direct-band radiative recombination efficiently because of phonon participation in the radiative recombination. The change in the band due to the tensile strain applied to Ge is shown in Figure 1b. The tensile strain causes Γ, *L* and the HH band to fall and the LH band to rise; the decrease in Γ is larger than that in *L*. The electron bandgap filling mechanism of the *L* band is shown in Figure 1c. After n-type doping of Ge, the *L* band can be filled equal to the Γ band, which means that this type of Ge can realize direct-band radiative recombination.

In this paper, we used Corsslight’s PICS3D and Optiwave’s OptiFDTD software as simulating and designing software. Because Si and Ge materials have the same zincblende lattice structure, we used the strained three-band zincblende model for gain and the radiative recombination method [31,32] as our carrier recombination approximate calculation model in the case of dislocation. The zincblende three-band model is derived from the 6 × 6 k.p model assuming parabolic bands. The model accounts for light (LHs), heavy (HHs) and split-off holes and for the effects of strain on carrier recombination.

The conduction, heavy hole, light hole, and split-off hole band-edge energies are calculated by the following equations:(1)$$\begin{array}{c} E_{c}=E_{v}+E_{g}+a_{c}\left(\varepsilon_{xx}+\varepsilon_{yy}+\varepsilon_{zz}\right) \end{array}$$
(2)$$\begin{array}{c} E_{hh}=E_{v}-P_{\varepsilon }-Q_{\varepsilon } \end{array}$$
(3)$$\begin{array}{c} E_{lh}=E_{v}-P_{\varepsilon }+\frac{1}{2}\left[Q_{\varepsilon }-\Delta_{0}+\sqrt{(}\Delta_{0}^{2}+9Q_{\varepsilon }^{2}+2Q_{\varepsilon }\Delta_{0}\right)] \end{array}$$
(4)$$\begin{array}{c} E_{so}=E_{v}-P_{\varepsilon }-\frac{1}{2}\left[Q_{\varepsilon }-\Delta_{0}+\sqrt{(}\Delta_{0}^{2}+9Q_{\varepsilon }^{2}+2Q_{\varepsilon }\Delta_{0}\right)] \end{array}$$
where Δ${}_{0}$ is split-off energy, *P${}_{\varepsilon }$* is
(5)$$\begin{array}{c} P_{\varepsilon }=-a_{v}\left(\varepsilon_{xx}+\varepsilon_{yy}+\varepsilon_{zz}\right) \end{array}$$

*Q${}_{\varepsilon }$* is:(6)$$\begin{array}{c} Q_{\varepsilon }=-\frac{b}{2}\left(\varepsilon_{xx}+\varepsilon_{yy}-2\varepsilon_{zz}\right) \end{array}$$
where *a${}_{v}$* and *b* are the valence band hydrostatic deformation potentials and ε${}_{xx}$, ε${}_{yy}$, and ε${}_{zz}$ are the strain tensor.

The effective masses for the various bands are given by the following equations:(7)$$\begin{array}{c} m_{hh}^{z}=m_{0}/\left(\gamma_{1}-2\gamma_{2}\right) \end{array}$$
(8)$$\begin{array}{c} m_{hh}^{t}=m_{0}/\left(\gamma_{1}+\gamma_{2}\right) \end{array}$$
(9)$$\begin{array}{c} m_{lh}^{z}=m_{0}/\left(\gamma_{1}+2\gamma_{2}f_{+}\right) \end{array}$$
(10)$$\begin{array}{c} m_{lh}^{t}=m_{0}/\left(\gamma_{1}-\gamma_{2}f_{+}\right) \end{array}$$
(11)$$\begin{array}{c} m_{so}^{z}=m_{0}/\left(\gamma_{1}+2\gamma_{2}f_{-}\right) \end{array}$$
(12)$$\begin{array}{c} m_{so}^{t}=m_{0}/\left(\gamma_{1}-\gamma_{2}f_{-}\right) \end{array}$$
where *m${}_{0}$* is free electron mass,γ${}_{1}$ and γ${}_{2}$ are Luttinger parameter, *f* is the strain factor expressed as:(13)$$\begin{array}{c} f_{\pm }=\frac{2s\left[1+1.5\left(s-1\pm \sqrt{1+2s+9s^{2}}\right)\right]+6s^{2}}{0.75{\left(s-1\pm \sqrt{1+2s+9s^{2}}\right)}^{2}+s-1\pm \sqrt{1+2s+9s^{2}}-3s^{2}} \end{array}$$
where S = Q${}_{\varepsilon }$/Δ${}_{0}$.

In this paper, we used the Fermi–Dirac statistical model for carrier statistics. We also used the parabolic QW model to predict bound state energies in the QW for modeling band structure. The solutions to the bound state energies are achieved by solving the Schrodinger equation along discrete slices in the quantization direction. Effective masses and band-edge parameters are taken from the strained three-band zincblende model for gain and radiative recombination methods. The Shockley–Read–Hall (SRH) recombination model [33,34] is added to simulate phonon transition in group IV of indirect-bandgap semiconductor materials. Considering the effects of doping and temperature, we added the Auger recombination model [35]. Considering the spontaneous emission characteristics of a laser, we added a spontaneous emission model. For the propagation of the light field, we used the finite-difference time-domain (FDTD) method to solve the Maxwell equations for an electromagnetic field. We used the material carrier absorption and free carrier loss models to estimate laser losses [36,37]. In this paper, the horizontal grid spacing is set to 100 nm. In the longitudinal mesh division, the grid spacing in the QWs region is set to 5 nm, while the other regions are set to 50 nm. Table 1 shows the parameters of Ge and Si in the calculation.

The K-selection rule of transition requires that the momentum and energy of quantum system must be conserved in the process of optical recombination. This law is expressed as follows:(14)$$\begin{array}{c} \hbar (k_{p}-k_{c}+k_{v})=0 \end{array}$$
where ***k***${}_{c}$ is conduction band electron wave vector, ***k***${}_{v}$ is valence band electron wave vector, ***k***${}_{p}$ is the photon vector. Formula (Equation 14) shows a vertical transition process involving only electrons, holes and photons. From the point of view of quantum mechanics, this transition belongs to the first-order perturbation process and has a high transition probability. For indirect band gap semiconductor materials, the bottom of conduction band and the top of valence band correspond to different K spaces, and their transitions do not follow the momentum conservation process shown in Formula (Equation 14), but there are phonons participating in the transition process of quantization of lattice thermal vibration energy, which makes the K selection rule still hold, the momentum conservation change to:(15)$$\begin{array}{c} \hbar (k_{p}-k_{c}+k_{v}\pm k_{s})=0 \end{array}$$
where ***k***${}_{s}$ is the wave vector of phonon. There are four kinds of quantum participation in this transition process, which belongs to the second-order perturbation process.

The Hamiltonian of the interaction between radiation field and electron in semiconductor is:(16)$$\begin{array}{c} \begin{array}{c} H=\left(\frac{-ae}{m_{0}}\right){\left(\frac{2\hbar }{\varepsilon_{0}n^{2}\omega }\right)}^{\frac{1}{2}}exp\left[j\left(k.r-\omega t\right)\right]P \end{array} \end{array}$$
where **P** = *-jℏ***∇** is momentum operator. We set the initial state of transition to be represented by conduction band electron wave function Ψ${}_{2}$ and the final state is expressed by valence band hole wave function Ψ${}_{1}$, the formula normalized to volume *V* expressed as:(17)$$\begin{array}{c} \begin{array}{c} \Psi_{2}\left(r,t\right)=V^{-\frac{1}{2}}u_{2}\left(r\right)exp\left[\left(k_{c}.r-\omega_{2}t\right)\right] \end{array} \end{array}$$
(18)$$\begin{array}{c} \begin{array}{c} \Psi_{1}\left(r,t\right)=V^{-\frac{1}{2}}u_{1}\left(r\right)exp\left[\left(k_{c}.r-\omega_{1}t\right)\right] \end{array} \end{array}$$
where *u(****r****)* is Bloch function of periodic characteristics of reaction lattice. Taking Formulas (Equation 16)–(Equation 18) into the Schrodinger equation, the conduction band electron transition probability **B${}_{21}$** is obtained, which expressed as: (19)$$\begin{array}{c} \begin{array}{c} B_{21}=\frac{\pi }{2\hbar }|<\Psi_{2}^{*}\left(r,t\right)\left|H\right|\Psi_{1}\left(r,t\right){>|}^{2} \end{array} \end{array}$$

Taking Formulas (Equation 16)–(Equation 18) and the Schrodinger equation into Formula (Equation 19), and in parallel taking the polarization direction of light to **k${}_{v}$**, we can conclude the conduction band electron transition probability is:(20)$$\begin{array}{c} \begin{array}{cc} B_{21}= & \frac{\pi e^{2}}{m_{0}^{2}\varepsilon_{0}n^{2}\omega }|<\frac{h}{2\pi j}V^{-1}\exp\left[j\left(\omega_{2}-\omega_{1}-\omega \right)t\right]\\ & \;\exp\left[j\left(k_{P}-k_{C}+k_{v}\pm k_{s}\right)r\right]u_{2}^{*}\left(r\right)\left(jk_{v}+\nabla \right)u_{1}\left(r\right){>|}^{2} \end{array} \end{array}$$

*u${}_{1}$*(***r***) is conduction band electron Bloch function, *u${}_{2}$*(***r***) is valence band hole Bloch function. When the light radiation field resonates with electrons in the semiconductor, then *ω${}_{2}$−ω${}_{1}-\omega$* = 0, the exponential function in becomes 1. In addition, we can see that when the direct band gap transition K selection rule in Formula (Equation 15) is satisfied, then ***k***${}_{p}$−***k***${}_{c}$ + ***k***${}_{v}$ ± ***k***${}_{s}$ = 0, then the exponential becomes 1 and Formula (Equation 20) has maximum value. It is theoretically verified that the direct bandgap transition process has the maximum phototransition efficiency.

The light emission of the TE mode is primarily due to the transition from conduction band to HH sub-band, whereas that of the TM mode is primarily due to the transition from conduction band to LH sub-band [20]. Therefore, according to the shortest transition criterion, the primary laser energy is concentrated on the TM mode. Otherwise, the width of optical transition band gap in QW cannot be determined by the band edge but is mainly determined by the ground state energy level located at the lower of valence band and the upper level of the conduction band. We calculated the transition bandgap energy diagram for different Ge layer widths by ground state energy level using a self-consistent Poisson-Schrodinger solver to select an appropriate Ge layer width as the active region of our laser, and the evaluation principle easily realizes the strain stretching process and makes the QW transition bandgap close to 0.8 eV. The result is shown in Figure 2a. As the thickness of the Ge layer increases, the transition bandgap clearly decreases. When the thickness of the Ge layer exceeds 35 nm, the decreasing tendency of the bandgap remains almost unchanged. To achieve a balance in terms of light wavelength and strain realization, we set the QW layer of the Ge laser to 35 nm-a QW thickness that is too large is not conducive to the realization of strain, as the lattice relaxation increases with the thickness.

To determine the suitable doping level of Ge, we obtained the laser EL intensity spectra of a nonstressed QW structure with different doping levels, which can reflect the *L*-band filling level, at room temperature. The result is shown in Figure 2b. As the n-doping level increases from 1 × 10${}^{16}$ cm${}^{-3}$ to 1 × 10${}^{19}$ cm${}^{-3}$, the EL intensity increases, and the peak energy gradually approaches 0.8 eV, which reflects the gradual filling of the *L* band with electrons. However, with a further increment in doping, EL intensity decreases sharply, mainly due to Auger recombination and carrier absorption. Auger recombination will greatly reduces the lifetime of minority carriers. To specify a suitable n-doping level to obtain the maximum light power, we choose 1 × 10${}^{19}$ cm${}^{-3}$ as our Ge QW doping level. It is necessary for a laser to maintain single-longitudinal-mode output to reduce unnecessary interference and loss. Therefore, a discussion regarding mode suppression approaches in the laser design is required. If the saturated output power of the main mode is strong enough and the other submodes are weak enough, we assume that single-longitudinal-mode operation can be achieved. The photon density of the main mode is:(21)$$\begin{array}{c} S_{0}=\frac{\gamma N/\tau_{s}}{\left(\frac{c}{n}\right)\left[\left(\alpha_{c}/\Gamma \right)-g_{p}\right]} \end{array}$$

The photon density of the other longitudinal modes is:(22)$$\begin{array}{c} S_{q}=\frac{\gamma N/\tau_{s}}{\left(\frac{c}{n}\right)\left[\left(\alpha_{c}/\Gamma \right)-g_{p}+{\left(\frac{q{\lambda_{q}}^{2}}{2Ln_{g}G_{0}}\right)}^{2}\right]} \end{array}$$
where α${}_{c}$ is the cavity loss, τ${}_{s}$ is the spontaneous emission recombination life, Γ is the light field limiting factor, *G${}_{0}$* is a parabola fitting factor, and γ is the spontaneous emission factor defined as the ratio of the spontaneous emission rate into each cavity mode to the total spontaneous emission rate.

γ is expressed as:(23)$$\begin{array}{c} \gamma =\frac{\lambda^{4}k}{8\pi^{2}n^{2}\Delta \lambda Vn_{g}} \end{array}$$
where *n${}_{g}$* is the group index of refraction, *V = dWL* is the active region volume, *L* is cavity length and Δλ is the width of the spontaneous emission spectrum. As seen from Equations (Equation 22) and (Equation 23), if the photon density of the other longitudinal modes is reduced to achieve single-longitudinal-mode output, the length of the cavity needs to be reduced and spontaneous emission factor γ increased therefore. This is the method that is adopted in this paper. However, as seen from Equation (Equation 21), reducing the cavity length will also increase the photon density of the main mode.

To explore suitable cavity lengths, we obtained the ratio of the submode power to the main mode power for different cavity lengths from 0 μm to 300 μm, with an interval of 25 μm. As shown in Figure 2c, when the cavity length is less than 100 μm, the submode is cleared, and the laser can realize single-longitudinal-mode operation. To maximize the output power, we choose a cavity length of 100 μm. Considering the spontaneous thermal characteristics of the device, we designed a unilateral negative electrode structure to provide the device with sufficient heat dissipation space. To improve the carrier injection efficiency and form a good spot shape, we designed a ridge waveguide structure confined by SiO${}_{2}$ in the upper-layer material.

As for the device process simulation, the Si substrate is monocrystalline silicon with the (100) crystal surface. Silicon materials are all deposited by CVD at 608 K. After that, the lower Si material was deposited at 608 K, and P element with a concentration of 1 × 10${}^{18}$ cm${}^{-3}$ was implanted by vertical ion implantation, with an energy of 150 kV. After the growth of the lower Si material, the Ge QW and Si barrier are alternately grown. The growth temperature of Ge QW layer is 1198 K and the 1 × 10${}^{19}$ cm${}^{-3}$ P element is doped by thermal diffusion meanwhile. The growth temperature of Si barrier layer is 598 K. Due to the thermal expansion mismatch between the Ge epitaxial layer and the Si substrate, 0.20% uniaxial tensile strain was introduced into the Ge layer. After the growth of active region, a 0.41 μm Si layer was deposited for the growth of oxide layer. Then the silica layer is oxidized using a wet oxidation process, which takes 300 min in an environment of 1173 K, 2 atm. Through the above steps, 0.95 μm oxide layer was successfully constructed. At last, we etched the electronic conductive channel filled with Si element by Si filling process, the filled Si is doped with 1 × 10${}^{18}$ cm${}^{-3}$ B element by thermal diffusion process. Finally, the left side window of the device is defined by etching process to form the cathode growth platform. Aluminum was used as the metal contact.

The distribution of strain in the laser is shown in Figure 2d as the output of the three-dimensional finite element method (3D-FEM) simulation solver. It can be seen from Figure 2d that the QW introduces a tensional strain of 0.2% in the XX direction. The three-dimensional structure of the side-emitting laser based on the Ge/Si QW structure is shown in Figure 2e. The laser possesses a cavity length of 100 μm. SiO${}_{2}$ occupies a 0.95-μm-thick and 2.5-μm-wide layer. The 1 × 10${}^{18}$ cm${}^{-3}$ p-type doped upper Si layer is 0.95-μm thick and 1-μm wide. Three 1 × 10${}^{19}$ cm${}^{-3}$ n-type doped Ge wells, with 0.2% tensile strain, are each a 35-nm thick and 6-μm wide layer, with two 0.2% compressive strain Si barriers as 50-nm-thick and 6-μm-wide layers consisting of QWs. The 1 × 10${}^{18}$ cm${}^{-3}$ n-type doped Si substrate consists of a thickness of 0.65 μm and a width of 6 μm. The cathode has a width of 4 μm and is 0.2 μm above the substrate.

## 3. Simulation Results

The light emitting field, is shown in Figure 3a. The emitted light is parallel to the material in the active region and forms a good elliptical light spot. The loss of the laser is shown in Figure 3b. The mirror loss mainly due to the defect loss of the mirror caused by the dislocation interface characteristics of the heterojunction and cavity. In addition to the mirror loss, the laser loss also caused by the carrier motion, and the loss of the optical absorption caused by the material, they cause carrier losses together. We also obtained the carrier loss in Figure 3b. When the threshold voltage is 0.8 V, the carrier loss of the device is 41.458 cm${}^{-1}$. With increases in the voltage, the carrier loss increases slowly. When the threshold voltage is 1.5 V, the carrier loss of the device is 48.712 cm${}^{-1}$. The mirror loss of the laser, which is 10.536 cm${}^{-1}$, is independent of the voltage. The band edge energy diagram of our Ge/Si QWs under a strain of 0.2% is shown in Figure 3c, obtained using the self-consistent Poisson-Schrodinger solver with deformation potentials. With the 0.2% tensile strain in the Ge layer, the Γ-band energy is 0.1722 eV, and the band energy of *L* is 0.0584 eV. The gap between Γ and *L* is 0.1138 eV, which is 0.0222 eV less than that of the unstrained Ge layer, indicating that the 0.2% tensile strain reduces the bandgap by 0.0222 eV. The LH band energy is −0.5991 eV, and the HH band energy is −0.6176 eV. The 0.2% tensile strain produces an energy split between the LH and HH levels of 0.0185 eV. The gains at the threshold voltage and 1.5 V are shown in Figure 3d. At the threshold voltage, the transverse electric (TE) mode gain reaches 180 cm${}^{-1}$, and the transverse magnetic (TM) mode gain reaches 600 cm${}^{-1}$. At 1.5 V, the TE mode gain reaches its peak at approximately 0.8 eV and 320 cm${}^{-1}$, whereas TM mode gain reaches its peak at approximately 0.8 eV and 930 cm${}^{-1}$. We also showed the TM gain distribution diagram for the whole device. The light emission of the TE mode is primarily due to the transition from conduction band to HH sub-band, whereas that of the TM mode is primarily due to the transition from conduction band to LH sub-band. We can see that the light gain in the QW region near the negative electrode is large, which is caused by the large current distribution near the negative electrode. The EL intensity spectrum result for the device are shown in Figure 3e. The EL intensity of the device near 1534 nm represents the peak value, which basically meets the requirement of light wave excitation at approximately 1550 nm. The width above half EL density is approximately 135 nm. By comparing Figure 3c,e, it can be found that the band gap determined by the band edge diagram of the laser is smaller than the actual transition energy, which is mainly caused by the insufficient strain introduction and the inconsistency between the first quantified energies and the band edge energy.

The relationship between the laser current and emitted laser power is shown in Figure 4a. The threshold current of the laser is approximately 3 mA. With a 300-mA current, the laser power is approximately 2.32 mW. The relationship between the laser voltage and emitted laser power is shown in Figure 4b. Figure 4b shows that the threshold voltage of the laser is 0.8 V. With a 1.5-V bias, the laser power is 2.32 mW, corresponding to a current of 300 mA. The gain confinement factor Γ defines the ratio of the transmitted power in the waveguide to the total power, which can measure the light leakage degree of the laser substrate. We obtained the gain confinement factor values of the laser at different voltages, as shown in Figure 4c. The gain confinement factor increases slightly with the voltage, which is mainly caused by the light emission and the diffusion caused by the increase in voltage. The diagram of the temperature change for different currents is shown in Figure 4d, obtained using the lattice heat flow equation solver. When the current is 300 mA, the temperature of the device reaches 340 K. The thermal distribution of the whole device at a 1.5-V operating voltage is shown in the appended diagram. The high-heating area of the device is close to the negative pole side. Because of the designed single-sided structure of the cathode, the heat can dissipate into the air at the upper side of the cathode in a timely manner, which is conducive to maintaining working stability. The diagram of the temperature change for different peak wavelengths is shown in Figure 4e. As the temperature increases, the peak wavelength gradually redshifts. This change is mainly due to the direct bandgap decreasing with increasing temperature. When the temperature is below 275 K, the peak wavelength plummets. At room temperature (300 K), the peak wavelength is approximately 1550 nm, which is in line with the device at room temperature for the 1550-nm luminescence wavelength requirement. Thus, the range of the laser operating temperature indicates that changes in the temperature of the laser will affect the peak wavelength; therefore, the peak wavelength can be turned by the laser temperature. The luminous powers for different temperatures is shown in Figure 4f. The maximum luminous power occurs at 300 K before 150 mA and at 280 K after 150 mA. When the temperature is greater than 300 K, the luminous power decreases with increasing temperature. This trend mainly reflects that the higher the temperature is, the more evident the heat loss of the carriers. When the temperature is lower than 260 K, the luminous power reduces to 0. This effect is mainly attributed to the decrease in the direct bandgap Γ point with the increase in the temperature and with the original doping concentration being insufficient to fill the gap between direct and indirect bands.

Figure 5a shows the light field diagram of five QWs. Based on the comparison between Figure 5a and Figure 3a, we can conclude that the light spot centre of the laser always tends towards the position of the most middle QW area. Because the conduction carriers entering the active region come from the Si barrier material and Si conduction band edge is associated to the valley, we should discuss the predicted gain value influence of barrier scattering processes. We calculated the normalized gain under different Si barrier widths to discuss the gain effect caused by carrier scattering in the barrier. The result is shown in Figure 5b. We can see that when the barrier width is less than 10 nm, the normalized gain increases with the increasing of the barrier. The main reason is that the small barrier width will be unfavourable to realize the carrier limit, which will result in the reduction of power. As the barrier width increasing, the barrier can effectively restrict the carrier, and the electronic wave function will not penetrate outside of the potential well. The normalized gain reaches its maximum when the barrier width is 10 nm and then decreases as the barrier width increases, the phenomenon is mainly because a too wide barrier will scatter carriers, which is not conducive to the gain increase. We choose the 50 nm barrier as the laser Si barrier width mainly because the 50 nm width of the barrier is conducive to the implementation of the Ge tension strain process. Figure 5c shows laser power of different QW numbers. The laser power of three QWs, four QWs and five QWs at a 1.5-V voltage is 2.32 mW, 2.66 mW and 2.79 mW respectively. The laser power difference between three QWs and four QWs is 0.34 mW, while the laser power difference between four QWs and five QWs is 0.13 mW. The increasing of laser power tended to moderate with the increasing of QW number which is mainly caused by the increased carrier dynamics loss caused by the increasing of QW number. Figure 5d shows the carrier loss with different QW numbers. The carrier losses of three QWs, four QWs and five QWs at a 1.5-V are 48.712 cm${}^{-1}$, 64.545 cm${}^{-1}$ and 77.466 cm${}^{-1}$, respectively. Carrier loss increase with the increasing of the numbers of QWs, which is mainly caused by the excessed carrier compound dynamics, that is, the more QWs there are, the more intense the radiation recombination interference between QWs. In addition, the scattering effect caused by the carrier recombination process of the upper QW will inevitably affect the radiation recombination efficiency of the lower QW, so the excessed carrier recombination effect has a great negative effect on the material gain. Figure 5e shows the gain confinement factors of different QW numbers. The gain confinement factors at a 1.5-V voltage of three QWs, four QWs and five QWs numbers are 0.1045, 0.1599 and 0.2042, respectively. We can see that with increases in the number of QWs, the gain confinement factor increases gradually, which is mainly caused by the increased number and area of QW, there is no doubt that this is beneficial to increase the laser power of the device, but as mentioned above, increasing the number of QWs will inevitably lead to the material gain decline, so the design of the number of QWs needs to balance the two relations. Figure 5f shows the material gain voltage under different QW numbers. In the TM mode at a 1.5-V voltage, three QWs, four QWs and five QWs gain are 930 cm${}^{-1}$, 659 cm${}^{-1}$ and 516 cm${}^{-1}$, respectively. In the TE mode at a 1.5-V voltage, three QWs, four QWs and five QWs gain are 320 cm${}^{-1}$, 180 cm${}^{-1}$, and 122 cm${}^{-1}$, respectively. The material gain decreases gradually as the number of QWs increases, which is also validates the previous discussion. Notably, the increasing of the number of QWs has no effect on the central wavelength of the laser. In conclusion, the increasing in the number of QWs will lead to the excessed carrier recombination effect and lead to the decrease in the material gain, but the increase in the number of QWs will increase the volume of the active region and also the gain confinement factor of the laser.

## 4. Conclusions

In summary, we introduced a group IV side-emitting laser with output at a wavelength of 1550 nm. We optimized the design of the Ge/Si QW and the device structure through calculations. When the Ge layer is 35 nm and the concentration of n-type doping is 1 × 10${}^{19}$ cm${}^{-3}$, the QW ensures the best stability of the strain structure and the maximum output power. When the laser cavity length is 100 μm and a single-side cathode structure is adopted, the laser realizes a single-longitudinal-mode emission and maintains thermal stability. At a 1.5-V voltage, the designed group IV side-emitting laser exhibits great edge-emitting characteristics and a high room temperature laser power at 1550 nm. The laser power can reach 2.32 mW, the gain of the TM mode is 930 cm${}^{-1}$, and the gain of the TE mode is 320 cm${}^{-1}$. We also compared the performance of three types of different QW lasers and found that increasing the number of QWs would increase the carrier loss of the laser, thus leading to the decrease of the gain of the material and slowing down the increase of the laser power. However, increasing the number of QWs will increase the gain confinement factor of the laser and increase the laser emission area, which is conducive to the improvement of power. This provides theoretical guidance for the subsequent design of QW lasers.

However, the growth temperature used in the Ge QW is relatively high. Although this is conducive to the realization of tensile strain, it will inevitably lead to the defects caused by the instability of Ge thin films. In the future work, it is necessary to balance the introduction of strain with the high-quality growth of materials. Secondly, the reliability of Ge QW laser also needs careful investigation, under so large current injection for a long time, the device materials is bound to be affected by high temperature and carrier mobility, which will damage the structure of the material, resulting in a decline in device performance. Thirdly, the exclusive packaging method of the laser also needs to be developed. In the process of packaging, it is necessary to avoid damage the laser and transmit as much laser power as possible to external photonic devices. Finally, the three band zincblende model adopted in the simulation should be corrected in order to make it more accurate for GeSi material simulation.

## Figures and Tables

**Figure 1 nanomaterials-10-01006-f001:**
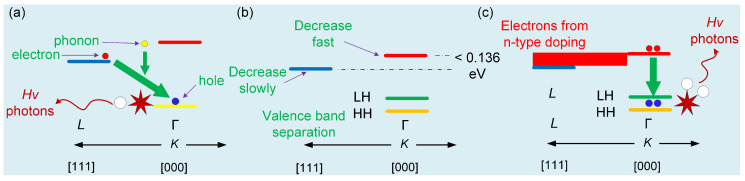
(**a**) Schematic band structure of an unstrained Ge well. (**b**) The schematic band structure of a tensile-strained Ge well. (**c**) The schematic band structure of a tensile strained Ge well via n-type doping.

**Figure 2 nanomaterials-10-01006-f002:**
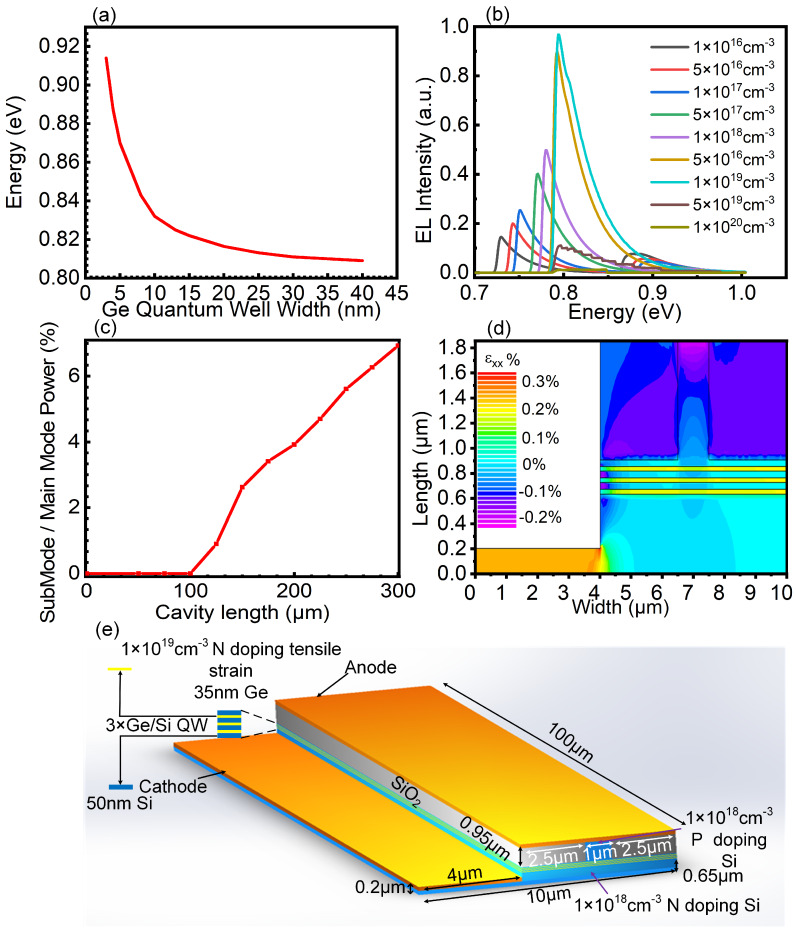
(**a**) Transition bandgap energy diagram determined by ground state energy level. (**b**) EL spectra for different doping levels. (**c**) Ratio of the emission power of the laser for different cavity lengths between the submode and the main mode. (**d**) The 3D-FEM simulation of the tensile-strained laser. (**e**) The schematic architecture of the designed laser 3D structure.

**Figure 3 nanomaterials-10-01006-f003:**
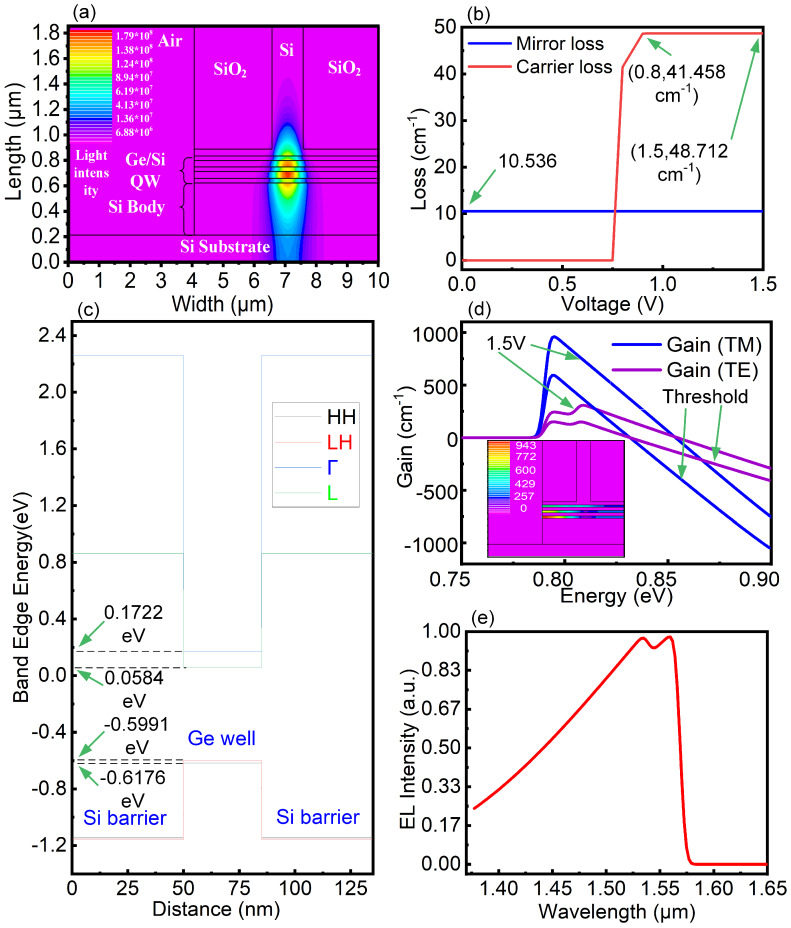
(**a**) Laser-light emitting field diagram. (**b**) The loss diagram of the laser. (**c**) The band edge energy diagram of the designed Ge/Si QWs under a strain of 0.2%. (**d**) Laser gain diagram at threshold and 1.5 V voltage. (**e**) The EL intensity spectrum of the laser.

**Figure 4 nanomaterials-10-01006-f004:**
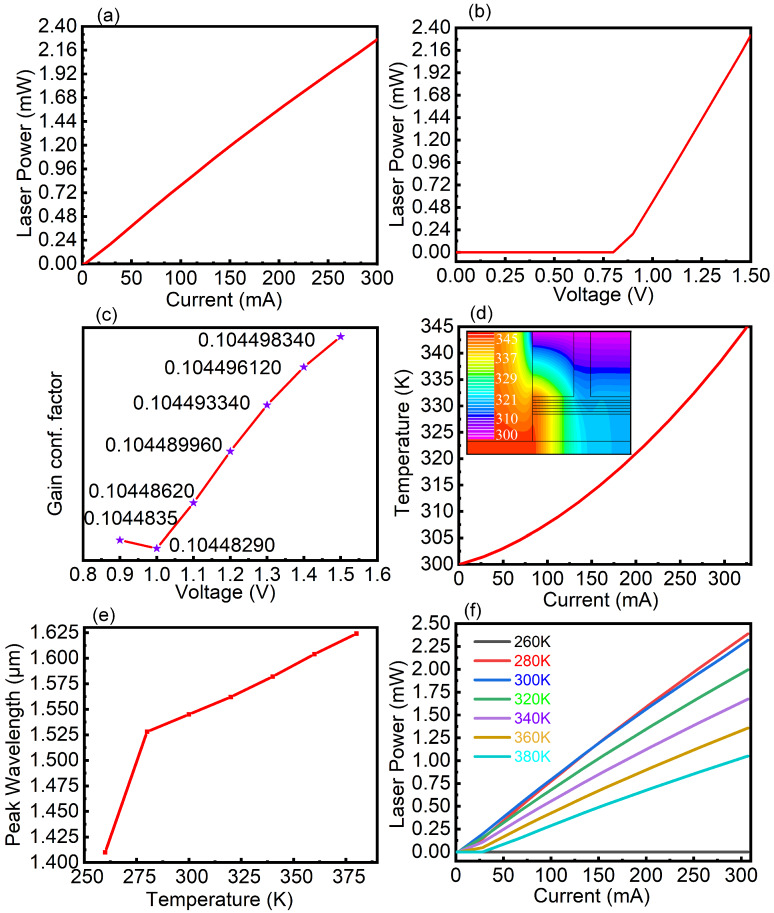
(**a**) The relationship between the laser current and laser power. (**b**) The relationship between the laser voltage and laser power. (**c**) The diagram of laser gain confinement factor. (**d**) The temperature change for different currents. (**e**) The temperature change for different peak wavelengths. (**f**) The luminous power diagram for different temperatures.

**Figure 5 nanomaterials-10-01006-f005:**
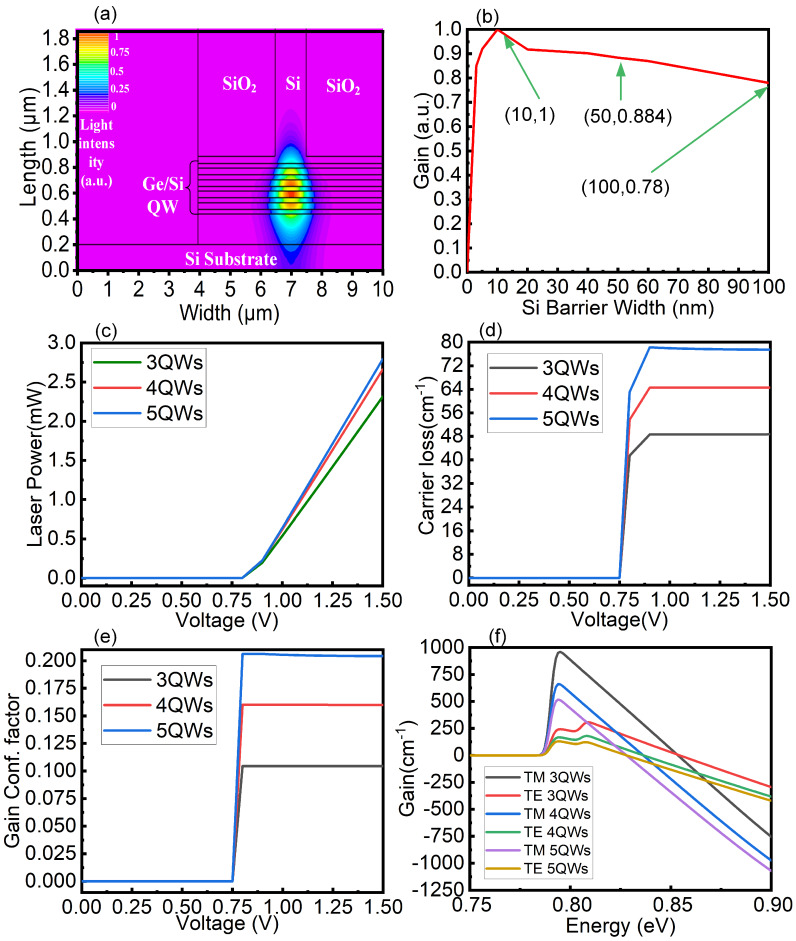
(**a**) Five Ge/Si QWs light field diagrams. (**b**) Normalized gain with different Si barrier widths. (**c**) Comparison diagram of laser power of different QW numbers. (**d**) Comparison diagram of carrier loss of different QW numbers. (**e**) Comparison diagram of gain confinement factor of different QW numbers. (**f**) Gain spectra of different QW numbers.

**Table 1 nanomaterials-10-01006-t001:** Material parameters for bulk Ge and Si at 300 K in our calculation.

Material Parameters	Si	Ge
SRH lifetime for electron and hole	$1\times 10^{-7}$ s	$1\times 10^{-7}$ s
Optical recombination rate	None	$1.5\times 10^{-11}$ cm${}^{3}$·s${}^{-1}$
Low field electron mobility	1350 cm${}^{2}$.V${}^{-1}$·s${}^{-1}$	3800 cm${}^{2}$.V${}^{-1}$·s${}^{-1}$
Low field hole mobility	500 cm${}^{2}$.V${}^{-1}$·s${}^{-1}$	1850 cm${}^{2}$.V${}^{-1}$·s${}^{-1}$
Isotropic dielectric constant	11.9	16.2
Thermal expansion coefficient	$2.59\times 10^{-6}$.K${}^{-1}$	$5.5\times 10^{-6}$.K${}^{-1}$
Effective masse$\left(m_{c}/m_{0}\right)$	0.156	0.038
Effective masse$\left(m_{t,L}/m_{0}\right)$	0.130	0.0807
Effective masse$\left(m_{l,L}/m_{0}\right)$	1.420	1.57
Effective masse$\left(m_{LH}/m_{0}\right)$	0.153	0.0424
Effective masse$\left(m_{HH}/m_{0}\right)$	0.537	0.316
Spin-orbit split off energy	0.044	0.29
Elastic constant $C_{11}$ (298 K)	165.77	128.53
Elastic constant $C_{12}$ (298 K)	63.93	48.26
Deformation potential $\alpha_{c}$	1.98 eV	−8.24 eV
Deformation potential $\alpha_{L}$	−0.66 eV	−1.54 eV
Deformation potential $\alpha_{v}$	2.46 eV	1.24 eV
Deformation potential *b*	−2.1 eV	−2.9 eV
Luttinger parameter γ${}_{1}$	4.34	12.93
Luttinger parameter γ${}_{2}$	0.33	4.05

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
