# Peer review of "The Structural, Electronic, and Optical Properties of Ge/Si Quantum Wells: Lasing at a Wavelength of 1550 nm"

_nanomaterials, 2020, doi:10.3390/nano10051006_

Round 1

Reviewer 1 Report

You are presenting primarily a theoretical study of an interesting concept without  obviously any full experimental verification like what is the real performance of the laser. However, your modelling is convincing and represents an invitation for technologists to embark on a full technological realization with  studies of power and noise behaviour.

Your paper would improve, if you can assess the technological problems, which need to be expected. Where are particular critical technology issues?

Can one expect reliability problems?

Author Response

The Structural, Electronic, and Optical Properties of Ge/Si QuantumWells: Lasing at a Wavelength of 1550 nm and Feedback

Hongqiang Li, Jianing Wang, Jinjun Bai, Shanshan Zhang, Sai Zhang, Yaqiang Sun, Qianzhi Dou, Mingjun Ding, Youxi Wang, Dan Qu, Jilin Du, Chunxiao Tang, Enbang Li, Joan Daniel Prades

Manuscript ID: nanomaterials-812064

Response to Reviewers

Thank you for your valuable suggestions. We have revised the paper according to your suggestions. Please note that the page numbers given below relate to the annotated version of the revised PDF.

Reviewer

  1. Comment –You are presenting primarily a theoretical study of an interesting concept without obviously any full experimental verification like what is the real performance of the laser. However, your modelling is convincing and represents an invitation for technologists to embark on a full technological realization with studies of power and noise behaviour.

Your paper would improve, if you can assess the technological problems, which need to be expected. Where are particular critical technology issues?

Can one expect reliability problems?

Authors’ response

Thank you for your valuable advice. Our ongoing work is to grow the proposed Ge QW laser and test the performance of it. We have supplemented the assessment of technological problems in the conclusion part of the paper from the aspects of QW growth temperature, device reliability, packaging and simulation model correction.

Action taken:

The assessment of technological problems have been supplemented on page 14, lines 334-343.

Additional revision

In order to improve the readability of the paper, we supplemented additional contents after revising the comments of reviewers. In Fig3(b) and Fig3(c) on page 10, we annotated the data of figure to make it correspond to the content discussed in the paper in order to improve the readability.

Action taken:

Data annotation of Fig.3(b) and Fig.3(c) has been supplemented on page 10.

Reviewer 2 Report

This manuscript presents a design and simulations of a side-emitting laser at telecom wavelength based on group IV semiconductors (Ge/Si). The authors optimize the design based on commercial software simulations of the laser output mode profile, wavelength tuning and lasing efficiency. Output powers at the level of 2 mW at 1550 nm are predicted at room temperature with a 300 mA current.

There is great interest (and challenge) in realizing efficient lasers using group IV materials which are compatible with silicon-based CMOS process. The authors proposes to realize this using a 3 layer quantum well design using Ge/Si and increase the direct-band radiative recombination by tuning the tensile strain of the Ge layer.

The research in group IV semiconductor lasers is timely and of interest for a journal like Nanomaterials and the design of a practical design and evaluation of its performances such as the one presented in this manuscript would be suitable for publication in Nanomaterials.

I have a few minor issues that the authors could address in order to improve the readability of their manuscript.
First, it would be easier to introduce the relevant levels, etc. of Fig. 1 in the beginning of section 2 before laying out the model used for the calculations.
Second, the authors refer to their numerical simulation result section as “Experiment”. This is misleading as the laser designed by the authors only exists on paper. The same goes for the proposed fabrication discussed on page 7, which is not actually realized. The authors should in general be clearer about the fact that their work is theoretical.
On page 7, after equation 22, there is some confusion about the various Gammas introduced. I also don’t understand why the authors write that “the spontaneous emission factor Gamma needs to be reduced, which can be achieved by reducing the cavity length”.

Author Response

The Structural, Electronic, and Optical Properties of Ge/Si QuantumWells: Lasing at a Wavelength of 1550 nm and Feedback

Hongqiang Li, Jianing Wang, Jinjun Bai, Shanshan Zhang, Sai Zhang, Yaqiang Sun, Qianzhi Dou, Mingjun Ding, Youxi Wang, Dan Qu, Jilin Du, Chunxiao Tang, Enbang Li, Joan Daniel Prades

Manuscript ID: nanomaterials-812064

Response to Reviewers

Thank you for your valuable suggestions. We have revised the paper according to your suggestions. Please note that the page numbers given below relate to the annotated version of the revised PDF.

Reviewer

  1. Comment – This manuscript presents a design and simulations of a side-emitting laser at telecom wavelength based on group IV semiconductors (Ge/Si). The authors optimize the design based on commercial software simulations of the laser output mode profile, wavelength tuning and lasing efficiency. Output powers at the level of 2 mW at 1550 nm are predicted at room temperature with a 300 mA current.

There is great interest (and challenge) in realizing efficient lasers using group IV materials which are compatible with silicon-based CMOS process. The authors proposes to realize this using a 3 layer quantum well design using Ge/Si and increase the direct-band radiative recombination by tuning the tensile strain of the Ge layer.

The research in group IV semiconductor lasers is timely and of interest for a journal like Nanomaterials and the design of a practical design and evaluation of its performances such as the one presented in this manuscript would be suitable for publication in Nanomaterials.

I have a few minor issues that the authors could address in order to improve the readability of their manuscript.

First, it would be easier to introduce the relevant levels, etc. of Fig. 1 in the beginning of section 2 before laying out the model used for the calculations.

Action taken:

The contents of Fig. 1 have been placed to the beginning of section 2 on page 3, lines 70-79. The models used for the calculations have been placed behind Fig. 1.

  1. Comment – Second, the authors refer to their numerical simulation result section as “Experiment”. This is misleading as the laser designed by the authors only exists on paper. The same goes for the proposed fabrication discussed on page 7, which is not actually realized. The authors should in general be clearer about the fact that their work is theoretical.

Action taken:

The word "Experiment" has been replaced with "Simulation results" on page 9, line 218. The discussions of fabrication have been revised on page 7, line 194.

  1. Comment – On page 7, after equation 22, there is some confusion about the various Gammas introduced. I also don’t understand why the authors write that “the spontaneous emission factor Gamma needs to be reduced, which can be achieved by reducing the cavity length”.

Authors’ response

Thank you for your valuable suggestions. The symbol of spontaneous emission factor is “gamma”. We have corrected it on page 7, lines 177 and 183. Please forgive our carelessness. As for your second question, longitudinal mode interval “delta lambda” at equation 22 is not expanded in the original paper, which contains the element of cavity length “L”, so the original description of the relationship between cavity length selection and edge mode suppression might be confusing. In the revised paper, equation 22 has been revised and the relationship between the spontaneous emission factor and cavity length has been redescribed. In the revised equation 22, the negative correlation between the photon density of the other longitudinal modes Sq and the cavity length L will be clearly shown.

Action taken:

We have corrected the sign “Gamma” to “gamma” on page 7, lines 177 and 183. equation 22 has been corrected, the longitudinal mode interval “delta lambda” in equation 22 has been expanded on page 7, line 175. The description of the interaction  between the spontaneous emission factor and cavity length has been revised on page 7, lines 183-185.

Additional revision

In order to improve the readability of the article, we supplemented additional contents after revising the comments of reviewers. In Fig3(b) and Fig3(c) on page 10, we annotated the data of figure to make it correspond to the content discussed in the paper in order to improve the readability.

Action taken:

Data annotation of Fig.3(b) and Fig.3(c) has been supplemented on page 10.